# Pluronic F127 and P104 Polymeric Micelles as Efficient Nanocarriers for Loading and Release of Single and Dual Antineoplastic Drugs

**DOI:** 10.3390/polym15102249

**Published:** 2023-05-10

**Authors:** Ramón A. Gutiérrez-Saucedo, Julio C. Gómez-López, Adrián A. Villanueva-Briseño, Antonio Topete, J. F. Armando Soltero-Martínez, Eduardo Mendizábal, Carlos F. Jasso-Gastinel, Pablo Taboada, Edgar B. Figueroa-Ochoa

**Affiliations:** 1Laboratorio de Proyectos Modulares, Departamento de Química, Centro Universitario de Ciencias Exactas e Ingeniería, Universidad de Guadalajara, Blvd. M. García Barragán 1421, Guadalajara 44430, Jalisco, Mexico; 2Laboratorio de Inmunología, Departamento de Fisiología, Centro Universitario de Ciencias de la Salud, Universidad de Guadalajara, Sierra Mojada 950, Guadalajara 44340, Jalisco, Mexico; 3Departamento de Ingeniería Química, Centro Universitario de Ciencias Exactas e Ingeniería, Universidad de Guadalajara, Blvd. M. García Barragán 1421, Guadalajara 44430, Jalisco, Mexico; 4Grupo de Física de Coloides y Polímeros, Departamento de Física de Partículas e Instituto de Materiales (IMATUS), Universidad de Santiago de Compostela, 15782 Santiago de Compostela, Spain

**Keywords:** Pluronic F127 and P104, drug release, doxorubicin and docetaxel, cervical cancer, pH response

## Abstract

The potential application of biodegradable and biocompatible polymeric micelles formed by Pluronic F127 and P104 as nanocarriers of the antineoplastic drugs docetaxel (DOCE) and doxorubicin (DOXO) is presented in this work. The release profile was carried out under sink conditions at 37 °C and analyzed using the Higuchi, Korsmeyer–Peppas, and Peppas–Sahlin diffusion models. The cell viability of HeLa cells was evaluated using the proliferation cell counting kit CCK-8 assay. The formed polymeric micelles solubilized significant amounts of DOCE and DOXO, and released them in a sustained manner for 48 h, with a release profile composed of an initial rapid release within the first 12 h followed by a much slower phase the end of the experiments. In addition, the release was faster under acidic conditions. The model that best fit the experimental data was the Korsmeyer–Peppas one and denoted a drug release dominated by Fickian diffusion. When HeLa cells were exposed for 48 h to DOXO and DOCE drugs loaded inside P104 and F127 micelles, they showed lower IC_50_ values than those reported by other researchers using polymeric nanoparticles, dendrimers or liposomes as alternative carriers, indicating that a lower drug concentration is needed to decrease cell viability by 50%.

## 1. Introduction

Hydrophobic drugs, i.e., drugs that are poorly soluble in water, are used for the treatment of various types of diseases, such as viral, bacterial and fungal infections, inflammation and cancer. Therefore, it is necessary to develop therapeutic materials at the nanoscale that optimize the solubilization process of the hydrophobic bioactive compounds in order to allow their sustained release and improve their pharmacokinetic profile and biodistribution to the target sites [1,2].

The development of polymeric materials capable of solubilizing, transporting, and releasing drugs has attracted great interest in the biomedical and pharmaceutical areas. In this context, nanocarriers that have already been examined for such a purpose include liposomes, dendrimers, polymeric micelles, nanogels, and other nanosized structures to improve charge, pharmacokinetics, biodistribution, and to reduce drug off-target toxicity [3,4]. Of particular interest are micellar structures formed due to the spontaneous self-assembly of hydrophobic blocks in aqueous solutions, generating a dense core with various morphologies such as spheres, rods and worm-like geometries, while the hydrophilic segments constitute the crown of the micelle [5]. Micelles are able to encapsulate non-polar drugs in the hydrophobic core, providing them with protection and stability, minimizing non-specific uptake, providing longer blood circulation times and driving the drugs to specific cells by the enhanced permeability and retention effect (EPR) in tumor tissues [6,7].

Block copolymers commercially known as linear poloxamers (Pluronic) are widely used in pharmaceutical preparations due to their biocompatibility, colloidal stability, and their ability to solubilize a wide variety of bioactive compounds thanks to the ability to form micellar structures in aqueous media. In this regard, conventional poloxamers [(PEO)_m_(PPO)_n_(PEO)_m_] are constituted by polyethylene oxide [(PEO), OCH_2_CH_2_] and polypropylene oxide [(PPO), OCH_2_CH(CH_3_)] chains in a triblock structure, where the subscripts m and n denote the average length of the corresponding block. Triblock poloxamers are amphiphilic, i.e., one part of their molecular structure is hydrophilic due to the PEO blocks, while the other is hydrophobic because of the PPO one [8], allowing the formation of self-assembling structures in solution at concentrations above their critical micellar value [9]. Moreover, they are thermosensitive materials; at temperatures below 15 °C, the solubility of these materials in water increases, and they exhibit a cloud point at elevated temperatures above 75 °C [8,10].

Poloxamers are listed in the U.S. and British Pharmacopoeia as excipients used in a wide variety of clinical applications [11], and are also approved by some regulatory agencies, such as the Food and Drug Administration (FDA) and the European Medicines Agency (EMA) for their use as food additives and pharmaceutical ingredients, and can also be used in pharmaceutical formulations and medical devices [12,13,14]. They have also been shown to modify the biological response sensitizing multidrug-resistant (MDR) cells and improving drug transport across cellular barriers [12,13].

Docetaxel (DOCE) and doxorubicin (DOXO) are antineoplastic drugs with an antibiotic structure, commonly used to treat various types of cancer including prostate, breast, lung and cervix. DOCE (C_43_H_53_NO_14_, 808.88 g·mol^−1^) is a taxane derivative that inhibits cell growth by stabilizing and preventing microtubule depolymerization [15]. However, due to its poor aqueous solubility, low bioavailability, and high toxicity, its clinical application is rather limited [16,17,18]. DOXO (C_27_H_29_NO_11_, 543.52 g·mol^−1^) is an antibiotic of the anthracycline family, which exerts its action by intercalating with the DNA double helix, preventing its replication and causing cell death. It has some drawbacks, such as cardiotoxicity, short half-life, and rather low aqueous solubility [19,20,21].

Recent studies have shown that it is possible to encapsulate these two drugs into polymeric nanocarriers and efficiently kill cancer cells, thus providing important synergistic effects while reducing acute toxicity and adverse side effects [7]. In addition, there are also some clinically approved nanoformulation-based micellar nanostructures. For example, Genexol, composed of paclitaxel-loaded micellar structures, is in Phase IIc clinical trials [22]; SP1049C (Supratek Pharma Inc., Montreal, Canada), a mixed micellar system composed of Pluronic L61 and F127, loaded with DOXO, has currently successfully undergone Phases I and II clinical trials [23]; and Taxatore, a formulation used by intravenous infusion, composed of Tween 80 and ethanol loaded with docetaxel [18]. In tumor interstitial fluids, spatial and temporal heterogeneities in blood flow lead to a metabolic microenvironment with significant pH variations.

This is a consequence of increased aerobic and anaerobic glycolysis. Thus, the pH that a drug carrier will face within the tumor microenvironment differs from the physiological pH. These local pH changes can be exploited to modulate the release of drugs encapsulated in nanocarriers [24,25,26].

In this regard, previous studies suggested that DOXO release could be controlled using the pH of the medium as an activator. For example, Liu et al. showed that under physiological conditions (pH 7.4), DOXO is released slowly from positively charged dendrimers, while at pH 5.5 (weak acidic tumor and endolysosomal microenvironment), the release rate increases rapidly [27]. Upadhyay et al. successfully demonstrated the encapsulation of DOCE in poly(γ-benzyl-l-glutamate)-block-hyaluronan polymersome PolyDOC system which released 20 and 40% of DOCE in a controlled manner in vitro at pHs 5.5 and 7.4, respectively, for the first 24 h [17]. Under physiological conditions, the hydrophobic interaction between the drug and the interior of the dendrimer is strong enough to retain the “dense core” conformation, which prevents drug efflux. When the environment changes to low pH, the conformation of the drug carrier changes from a “dense core” to a “dense shell” due to ionic pairing, which accelerates drug release [27].

Hence, in the present work, Pluronic P104 and F127 triblock copolymers were used to develop a therapeutic nanoplatform that acts as an efficient carrier for single and dual loading of the antineoplastic drugs DOCE and DOXO, in order to analyze the encapsulation efficiency, colloidal stability, release kinetic profile in a physiological medium at variable pH and cytotoxicity against the HeLa cervical cancer cell line. The findings of this work suggest that micellar solutions formed by Pluronic F127 and P104 can provide an attractive, effective and biocompatible platform for single and/or dual solubilization of the hydrophobic drugs DOCE and DOXO, providing pharmacologically synergistic chemotherapeutic combinations of sufficiently high potency to kill cancer cells. Thus, these platforms can be used in combination for chemotherapy to treat different types of cancer by parenteral administration, with minimal side effects associated with the vehicle.

## 2. Materials and Methods

### 2.1. Materials

Triblock copolymer Pluronic F127 (PEO_100_PPO_65_PEO_100_, Mw = 12.5 kDa), methyl alcohol anhydrous, ethyl alcohol, acetone, dimethyl sulfoxide (DMSO), anhydrous dichloromethane, phosphorus pentoxide, triethylamine, methylene chloride, and phosphate-buffered saline (PBS) were purchased from Sigma-Aldrich, and the Pluronic P104 (PEO_27_PPO_61_PEO_27_, Mw = 5.9 kDa) from BASF Chemical Company. Docetaxel (DOCE) and doxorubicin hydrochloride (DOXO·HCl) were acquired from PISA Laboratories. DOXO base (DOXO-B) was obtained by precipitation of aqueous DOXO·HCl solution (1 mg·mL^−1^) by the addition of triethylamine and methylene chloride. The drug was kept under vigorous stirring for one hour, and the organic phase was evaporated to recover the DOXO-B base [28]. HeLa cells, donated by Dr. Ana Laura Pereira (Laboratorio de Inmunología, Universidad de Guadalajara), were from ATCC. Eagle’s Minimum Essential Medium (EMEM), fetal bovine serum (FBS), trypsin-EDTA, phosphate-buffered saline (PBS), penicillin/streptomycin and cell counting kit-8 (CCK-8) were purchased from Sigma-Aldrich.

### 2.2. Drug Encapsulation

Drug loading in micellar solutions was achieved following the method reported by Elsabahy with some modifications [9]. This modified method consists of preparing solutions in dichloromethane (200 µg·mL^−1^) of the DOCE drugs (intrinsic solubility in water 0.274 mg·dm^−3^) [29] or DOXO-B (intrinsic solubility in water 0.5 mg·dm^−3^) [30]. Then 50 to 300 µL of these solutions are added to 20 mg of F127 or to 50 mg of P104. The mixture is kept under magnetic stirring at 250 rpm and room temperature of 25 °C, and then 1 mL of deionized water is added dropwise to form the micellar solution.

Once the drug-encapsulated micellar system is obtained, the vials are keep uncapped, under magnetic stirring at 250 rpm and room temperature of 25 °C for five days to evaporate the organic solvent. Then, the micellar solution is centrifuged at 3000 rpm for 30 min. The supernatant is filtered through 0.45 µm Millipore Millex cellulose membranes to discard the non-encapsulated drug [31].

Drug-loaded micellar solutions were prepared in triplicate and stored to protect them from light by coating the vials with aluminum foil to prevent the degradation of the drugs. Drug-loaded micellar solutions were also prepared using mixtures of DOCE and DOXO-B with weight percentages of 25:75, 50:50, and 75:25, respectively. The amount of drug in the micellar solutions was determined by ultraviolet–visible absorption spectrophotometry (UV–Visible Evolution 220 Thermo Scientific Spectrophotometer), diluting the filtered supernatant in methanol in a 1/1000 ratio. Previously, calibration curves for each drug in methanol were made. The wavelength used to determine the amount of drug was 227 nm for DOCE [9] and 480 nm for DOXO [21]. The experiments were performed in triplicate. The parameters: drug loading (*D.L.*), entrapment efficiency (*E.E.*), and solubilization capacity per gram of copolymer in solution (*S_cp_*) [32,33] were calculated using the following equations:(1)$$D.L.\left(\%\right)=\left[\frac{Weight\;of\;the\;drug\;in\;micellar\;solution}{Weight\;of\;polymer\;+\;drug}\right]\times 100\%$$
(2)$$E.E.\left(\%\right)=\left[\frac{Weight\;of\;the\;drug\;in\;micellar\;solution}{Weight\;of\;feeded\;drug}\right]\times 100\%$$
(3)$$S_{cp}=\left[\frac{Weight\;of\;the\;drug\;in\;micellar\;solution\;(mg)}{Weight\;of\;polymer\;(g)}\right]$$

### 2.3. Particle Size of Pure and Drug-Loaded Micellar Systems 

The micelle particle size measurements with and without drug were performed by dynamic light scattering (DLS) at an angle of 90° and temperature of 40 °C, using the Zetasizer Nano Z-S90 Light Scatterer from Malvern Instruments. The DLS correlation functions were analyzed by the CONTIN method to obtain the intensity distributions, the apparent diffusion coefficients and the apparent hydrodynamic radius (*r_h app_*), using the Stokes–Einstein equation [33]:(4)$$r_{h\;app}=\frac{kT}{6\pi \eta D_{app}}$$
where *k* is the Boltzman’s constant, *η* is the dynamic viscosity of water, *T* is the absolute temperature, and *D_app_* is the apparent diffusion coefficient. Before particle size measurements, the micellar solution alone or loaded with the drug was filtered through Millipore Millex cellulose membranes with 0.45 µm pore size and placed in a previously washed and dried glass cell; the system was stabilized inside the equipment at the measurement temperature (40 °C) for 10 min. The experiments were performed in triplicate.

### 2.4. In Vitro Drug Release

The release kinetic profile experiments of DOCE and DOXO from Pluronic F127 and P104 were performed under “sink” conditions in a PBS buffer medium at pH of 5.5 (simulated intestinal fluid) and 7.4 (simulated parenteral administration) at a temperature of 37 °C under constant stirring. To facilitate drug diffusion across the membrane and to avoid the formation of aggregates, 2 (*v*/*v*)% ethanol was added to the medium [17]. The micellar solutions (5 mL) loaded with the individual drug or drug combination were deposited inside a dialysis tube (Spectrapore, MWCO 3500 Da) and subsequently immersed in 500 mL of the release medium at the established conditions [17]. Aliquots of 1 mL of the release medium were taken at determined times, and the volume extracted from the medium was replaced with fresh PBS.

Subsequently, the drug concentration was quantified by the UV–Vis spectrophotometry technique. The experiments were carried out in triplicate. The experimental data on the release kinetic profile of DOCE and DOXO were fitted with three drug release models: Higuchi, Korsmeyer–Peppas and Peppas–Sahlin. Higuchi’s model describes drug release from an insoluble matrix as the square root of time and is based on Fick´s second law [34]:(5)$$\frac{M_{t}}{M_{\infty }}=kt^{1/2}$$

The Korsmeyer–Peppas model is a modification of the Higuchi model and is used to determine if more than one transport mechanism is involved [34,35]:(6)$$\frac{M_{t}}{M_{\infty }}=k_{1}t^{n}$$

The Peppas–Sahlin model consists of two terms: the first term is the Fickian contribution and the second one is the chain relaxation contribution [36,37]:(7)$$\frac{M_{t}}{M_{\infty }}=k_{2}t^{m}+\;k_{3}t^{2m}$$
where *M_t_* represents the amount of drug released at the time (*t*) and *M_∞_* the amount of drug loaded, *k* is the kinetic constant of the Higuchi model, *k*_1_ is the kinetic constant of the Korsmeyer–Peppas model and *n* is the Fickian diffusion exponent, *k*_2_ and *k*_3_ are the Fickian kinetic constant and the relaxational/dissolution rate constant of the Peppas–Sahlin model respectively, and *m* is the Fickian diffusion exponent. Fickian diffusion exponent values lower than 0.43 indicate a drug release profile controlled by classical diffusion through spherical structures; for values between 0.43 and 0.85, the drug release is of the anomalous type (a combination of classical diffusion with the relaxation mechanism of the membrane), associated with the tensions generated between the transport vehicle and the drug. Finally, when the values are greater than 0.85, the release of the drug is subject to the relaxation of the polymeric chains [36,38]. DDSolver software was used to fit the experimental data to the drug release kinetic models [39].

### 2.5. Cell Viability Assays

HeLa cells were seeded in a 75 cm^3^ flask with 10 mL of the complete medium containing 10% of FBS and 1% of penicillin/streptomycin; when cells were confluent, they were trypsinized and seeded in a 96-well plate at 5000 cells per well with 100 µL of complete medium and incubated at 37 °C with 5% CO_2_ for 24 h. Then, the culture medium was replaced with fresh medium containing micelles, and serial dilutions were made. Cells without micelles and cells treated with etoposide were used as 100% and 0% viability controls. Cells with different treatments were incubated at 37 °C with 5% CO_2_ for 48 h. Each treatment was evaluated in triplicate. Cytotoxicity, as a percentage of cell viability was determined by the CCK-8 assay, according to the manufacturer´s instructions.

After 48 h, medium was removed, cells were washed gently with PBS, and medium containing 10 (*v*/*v*)% of CCK-8 reagent was added to each well. Cells were incubated for 3 h at standard conditions and the absorbance at 450 nm was measured in a microplate reader (Multiskan Go, Thermo Scientific). Viability was calculated with the following equation:(8)$$\%\;viability=\frac{{Abs}_{sample}}{{Abs}_{control}}\times 100\%$$
where *Abs_sample_* is the absorbance of cells treated with micelles, and *Abs_control_* is the absorbance of cells without treatment (fresh medium). The IC_50_ of each treatment was obtained by fitting the experimental data to a three-parameter (inhibitor) versus response model using GraphPad Prism software.

## 3. Results

### 3.1. Drug-Loading Capacity of Polymeric Micelles

Two triblock copolymers with different amounts of hydrophilic building blocks in their molecular structure, Pluronic F127 (PEO_100_PPO_65_PEO_100_) and P104 (PEO_27_PPO_61_EO_27_), were used to prepare micellar solutions at 2 and 5 *wt*% in water, respectively; these concentrations are much higher than their respective critical micellar concentration (CMC), (0.3 mg·mL^−1^ for Pluronic F127 and 0.7 mg·mL^−1^ for Pluronic P104), ensuring complete micelle formation [40]. Because DOXO hydrochloride (DOXO-HCl) is commercially available and is originally hydrophilic, the hydrophobic DOXO base (DOXO-B) was obtained by adding triethylamine to the DOXO-HCl solution in a 3:1 molar ratio (triethylamine/DOXO-HCl). The mixture was kept under conditions of stirring at 500 rpm and 25 °C for 1 h; subsequently, dichloromethane was added to form a two phase solution (aqueous and organic) where DOXO-B was dissolved in the organic phase [28]. DOXO-B was finally obtained by the evaporation of dichloromethane.

Next, 10 to 60 µg of the drugs docetaxel and doxorubicin (base) were added to 1.00 g of each polymeric micellar solution. The amount of drug encapsulated within the respective Pluronic F127 and P104 micellar solutions was determined by UV–Vis spectrophotometry [9,31]. Single and dual (DOCE/DOXO-B) encapsulation experiments were carried out. Table 1 shows the amount of drug incorporated into the polymeric micellar solutions, reported as the percentage of the drug that is entrapped in the micelles (*D.L.*), the entrapment efficiency (*E.E.*), and the drug solubilization capacity per gram of copolymer (*S_cp_*). When the amount of drug added to the Pluronic F127 micellar solutions was increased, the encapsulated amount also increased until a saturation value was reached. A similar observation was made for Pluronic P104 micelles, but without reaching equilibrium at the maximum concentration used.

At saturation for the F127 micellar system, the amount of drug solubilized per gram of copolymer (*S_cp_*) was 1.424 mg·g^−1^ for DOCE and 1.145 mg·g^−1^ for DOXO-B. In addition, DOCE drug entrapment efficiency was 56.8% whereas for DOXO-B was 45.8%. For the P104 micellar system the drug solubilization capacity was 0.710 mg·g^−1^ for DOCE and 0.609 mg·g^−1^ for DOXO-B, while the encapsulation efficiency for DOCE was 88.5% and 76.0% for DOXO-B.

In addition, and as depicted in Table 1, the amount of *D.L.* seemed to reach a maximum value beyond which further enhancement no longer resulted in additional solubilized drug inside micelles and drug precipitation could occur; this is in agreement with previous reports using other block copolymers such as Pluronic P123 [41], some Tetronics [42], and some poly(styrene oxide)-poly(ethylene oxide) [43] and poly(butylene oxide)-polyethylene oxide [44]. When the same drug/copolymer mass ratio (D/C) in both polymeric micellar solutions was analyzed, for example, 0.10 D/C%, it was observed that the amount of drug solubilized was higher when Pluronic P104 was used, which may be favored by its higher proportion of hydrophobic groups giving rise to a more hydrophobic micellar core [45]. Furthermore, the amount of drug incorporated into Pluronic F127 and P104 micellar solutions was much higher than values previously reported regarding the water solubility of DOCE (0.274 mg·dm^−3^) [29] and DOXO (0.5 mg·dm^−3^) [30] and increased as the amount of polymer in solution was higher.

For example, Table 1 shows that when drugs were added to Pluronic F127 micellar solutions at a percentage of 0.10% with respect to the copolymer (D/C), the solubility of DOCE increased by a factor of 43 and that of DOXO-B by a factor of 21; for Pluronic P104 micelles, the solubility of DOCE increased by a factor of 147 and that of DOXO-B by a factor of 61. Due to their hydrophobic character, it is expected that most of the drug will be encapsulated in the micellar core and micellar core–corona interface, allowing it to be transported undiluted into the bloodstream [46]. Conversely, the hydrophilic micellar PEO shell should minimize drug uptake by the reticuloendothelial system, favoring its circulation time and accumulation in solid tumors [47].

Dual loading of DOCE/DOXO was carried out in the Pluronic F127 and P104 micellar systems, maintaining the total concentration of the added drugs at a constant. The DOCE/DOXO ratios used were 25:75, 50:50, and 75:25. Table 2 shows the amount of each drug encapsulated into the respective micellar solutions.

For dual drug loading inside the Pluronic F127 micellar solution, a D/C% ratio of 0.25 was used because in the encapsulation tests using single drugs, this ratio allowed the incorporation of the highest amount with a good entrapment efficiency; in contrast, for dual drug loading into Pluronic P104 micellar solution, a D/C% ratio of 0.08 was used for the same reason. Table 2 shows that when a 50:50 DOCE/DOXO ratio was used, a maximum entrapment efficiency of 49.2% for Pluronic F127 and 78.6% for Pluronic P104 was obtained, with a maximum solubilization capacity (*S_cp_*) of 1.231 mg·g^−1^ and 0.628 mg·g^−1^, respectively. In addition, for both micellar systems, it was observed that the entrapment efficiency slightly decreased when using the dual loading system compared to the single one.

For example, Villar-Álvarez et al. showed that in dual DOCE/DOXO-loaded BO_n_EO_m_BO_n_ micelles, the co-encapsulation of the two drugs inside the micellar core provided a slight increase in the whole *D.L.* capacity (ca. 1.26 *w*/*w*%) compared to the individually encapsulated compounds ca. 0.9 and 1.1 *w*/*w*% for DOCE and DOXO loading, respectively [7]. Shin et al. previously reported that the encapsulation of multiple anticancer drugs, such as paclitaxel, DOCE, and etoposide encapsulated into a PEG-b-PLA micellar system did not adversely affect the solubility of the individual drugs and provided favorable therapeutic synergistic effects in the fight against cancer [48].

In the same line, Berko et al. determined the cytotoxic effects on cancer cells of polymeric nanoparticles, carried out simultaneously by two chemotherapeutic drugs of complementary action, highlighting their enhanced therapeutic efficiency as a result of the synergistic action of both drugs [49]. Furthermore, Hasenstein et al. used a micellar system consisting of poly(ethylene glycol)-poly(lactic acid) capable of encapsulating three antineoplastic drugs (paclitaxel, rapamycin, and 17-AAG), providing a strong synergistic cytotoxic effect on MDA-MB-231 breast cancer cell lines [50].

### 3.2. Micellar Size of Drug-Loaded Polymeric System

The size measurements of DOCE, DOXO and DOCE/DOXO loaded in micelles of F127 and P104 triblock copolymers were performed by DLS at 40 °C; a temperature greater than their critical micellar temperature (CMT), to ensuring complete micelle formation [40]. All samples were filtered through cellulose membranes (0.45 µm) and stabilized at the established temperature for 10 min. The physical appearance of the micellar solutions was fluid, homogeneous and transparent. Figure 1 shows the particle size distribution of the bare and drug(s)-loaded Pluronic F127 and P104 micelles for which monodisperse and narrow population size distributions (polydispersity lower than 0.5) were observed.

The particle diameter of Pluronic F127 micelles alone at 40 °C was 24.4 nm. When loaded with 50 µg of the drug, it increased to 122.4 nm with the addition of DOXO-B (Figure 1a), and 43.8 nm when DOCE was used (Figure 1b). In the case of Pluronic P104, bare micellar size at 40 °C was 18.2 nm, but when loaded with 40 µg of the drug, it increased to 91.3 nm for DOXO-B (Figure 1c) and to 50.7 nm for DOCE (Figure 1d), respectively. In summary, it could be observed that the size distribution shifted to larger values when the drug was incorporated in the polymeric micelles, in agreement with previous works analyzing the potential of other types of micellar nanostructures to transport several antineoplastic and antifungal drugs [7,32,44].

In fact, the larger particle size of DOCE-loaded micelles compared to DOXO-B-loaded ones could be attributed to the hydrophobic interactions between the aromatic rings of DOCE within the micellar core and the hydrogen bonds and Van der Waals forces between the hydroxy groups of the drug and the oxygen of the block copolymers [15].

The dispersion stability of drug-loaded micelles is an essential factor for the evasion of detection and destruction by the reticuloendothelial system (RES). The stability of DOCE and DOXO loaded in the micelles of Pluronic F127 and P104 was determined by monitoring the micellar size evolution (Figure 2). In this figure, it can be observed that the size of the micelles increased in the first days, before it became fairly constant after the eleventh day. The larger increase was found for DOXO-loaded F127 of ca. 230 nm, while the smallest increase was for DOCE-loaded P104, ca. 132 nm.

Similar behavior was observed by Villar et al., who reported a size increase in DOCE/DOXO-loaded poly(butylene oxide)-poly(ethylene oxide)-poly(butylene oxide) polymeric micelles. The size increase occurs due to a structural rearrangement of the micelles, which may lead to the preclusion of some hydrophobic blocks outside the micellar core favoring the formation of intermicellar bridges [7].

It is also worth mentioning that the hydrodynamic diameter of Pluronic F127 and P104 polymeric micelles decreased with increasing DOCE/DOXO ratio (Figure 3), which is in agreement with the data shown in Figure 1, in which micelles loaded with DOCE were smaller than those with DOXO-B. In particular, for the DOCE/DOXO-loaded Pluronic F127 micellar system, sizes of ca. 91.3, 78.8 and 50.8 nm were observed for 75DOXO:25DOCE, 50DOXO:50DOCE and 25DOXO:75DOCE ratios, respectively; meanwhile, for dual-loaded Pluronic P104 micelles sizes of 78.8, 68.1 and 58.8 nm, respectively, were detected for the same DOXO/DOCE ratios. Villar et al. studied the size of BO_n_EO_m_BO_n_ triblock copolymer micelles obtained when loaded with DOCE/DOXO mixtures, and found relatively similar size increases compared to micelles loaded with a single drug [7].

The development of nanomaterials to be used as therapeutic platforms to combat some diseases is constantly evolving with attempts to generate new alternatives for the controlled release of multiple active ingredients in a single dosage system. Villar et al. found that single or dual chemotherapeutic drug-loaded polymeric micelles below 120 nm in size were viable to enable their tumor-specific accumulation via the enhanced permeability and retention (EPR) effect [7].

### 3.3. In Vitro Release Profile from Polymeric Micelles

The release profiles of DOCE and DOXO drugs from micellar solutions of 2 *wt*% Pluronic F127 and 5 *wt*% Pluronic P104 were obtained under sink conditions at 37 °C and continuous stirring at 200 rpm. Micellar solutions loaded with the drugs were introduced into a dialysis tube (Spectra Pore, 3500 Da cut-off cellulose ester membrane) immersed in PBS physiological buffer medium (pH of 5.5 and 7.4) in which 2 (*v*/*v*)% ethanol was added [17]. Previously, DOCE and DOXO calibration curves in PBS were obtained (not shown). Figure 4 shows the drug release profiles from Pluronic F127 and P104 micellar solutions in physiological mimicking media at pHs of 7.4 and 5.5. The in vitro release profiles were similar for all analyzed conditions.

When single drug release from polymeric micelles was carried out at pH 7.4 (Figure 4a), the drug release was relatively fast during the first 12 h followed by a slower release rate until the end of the experiments. For Pluronic F127 micellar solutions at 12 h, 55% of DOCE and 44% of DOXO were released, and after 48 h up to 88% of DOCE and 57% of DOXO were liberated from the micellar nanocarrier. When Pluronic P104 micellar solutions were used, after 12 h of incubation, 39% of DOCE and 40% of DOXO were released, and ca. 73% of DOCE and 72% of DOXO after 48 h. It was also observed that the amount of DOCE released from Pluronic F127 micelles was higher than that from Pluronic P104 ones, the opposite to what was observed for encapsulated DOXO. Previous studies have shown that the concentration of the micellar solution and the number of hydrophobic blocks in the copolymers are determining factors for drug retention within the micellar structure [44].

In contrast, the amount of DOCE and DOXO released at acidic pH 5.5 (Figure 4b) was greater than that at pH 7.4, and it was released faster. After 24 h of incubation, the percentages of drug released from the Pluronic F127 micellar system were 94% for DOCE and 99% for DOXO, and from the Pluronic P104 micellar system, they were 95% for DOCE and 94% for DOXO. These data can be expected because, at pH 5.5, the micelle shell is slightly protonated, making it more hydrophilic and favoring water intercalation to some extent, thus, facilitating the diffusion of the drug molecules [42].

Figure 5 shows the release profiles of 50%DOCE-50%DOXO loaded into Pluronic F127 and P104 micelles. As shown before, drug release was fast within the first 12 h, followed by slower rate up to the end of the experiments. Dual-loaded micelles also showed a faster drug release when the pH decreased. The in vitro release profiles of the single and double-loaded drug in polymeric micelles were analyzed with Higuchi, Korsmeyer–Peppas, and Peppas–Sahlin diffusion models [45]. Table 3, Table 4 and Table 5 show the parameters obtained from the fitting data. The Peppas–Sahlin diffusion model provided a priori the best fit for the experimental data with a correlation coefficient, R^2^, larger than 0.981. Additionally, this diffusion model showed the lowest values of the mean square error (MSE) and the largest values for the model selection criterion (MSC) [39].

For Pluronic F127 micellar solutions, an anomalous diffusion process was determined (*m* > 0.43) associated with the tensions between the drug and copolymer, while for the Pluronic P104 micellar system, values of *m* < 0.43, were indicative of a classic drug release profile with spherical structures. The constant associated with drug diffusion (*k_2_*) showed a similar value for all copolymers analyzed, for both single and dual-loaded micelles, which is indicative that a common mechanism of drug release was carried out in a similar way, regardless of the drug loaded inside the micellar solutions. On the other hand, the diffusion constant was affected when the pH immersion medium used was varied, specifically, increasing its value when the release profile in an acid medium (pH 5.5) was analyzed.

However, some *k_3_* values (related to chain relaxation) are negative, which do not have physical meaning, because in these cases the Fickian diffusion term gives cumulative release values greater than 100% for release times over 40 h. Furthermore, the *k_3_* values are much smaller than *k_2_* values, indicating that the contribution of the chain relaxation mechanism to the drug release is small. In this regard, the correlation coefficient R^2^ of the Korsmeyer–Peppas model is not very distant from that of the Peppas–Sahlin model, and when *k_3_* values are positive, the correlation coefficients R^2^ are similar.

When comparing the release curves obtained by both models (not shown), it was found that they were almost identical. Then, it was concluded that the drug release occurred by Fickian diffusion and that the Korsmeyer–Peppas model fitted correctly to the experimental data. It was reported that in the release of the drug Nicorandil from floating tablets using a combination of hydrophilic and hydrophobic polymers, the Korsmeyer–Peppas and the Peppas–Sahlin models fitted the experimental data well. However, in the Peppas–Sahlin model, much smaller values of *k_3_* than *k_2_* values were obtained, including in some cases negative *k_3_* values, which indicated that the relaxation mechanism was insignificant, concluding that the drug release mechanism was by Fickian diffusion [51]. Figure 4 and Figure 5 show the good agreement between the experimental values and the Korsmeyer–Peppas model.

### 3.4. Cell Viability Assays of Individual and Combined Micelles

The cell viability of HeLa cells exposed to different concentrations of empty, and DOXO-B- or DOCE-loaded micelles was evaluated using the proliferation cell counting kit CCK-8 assay, which is based on the metabolic activity of cells and their capacity to reduce the water-soluble tetrazolium salt WST-8 to formazan by cellular dehydrogenases. The IC_50_ (the concentration of drug needed to decrease the viability of cells to 50%) of the micelles was determined by fitting the data to an (inhibitor) vs. response three-parameter model. As can be observed in Figure 6a,b and Table 6, the empty micelles of F127 were practically non-cytotoxic (IC_50_ > 1 × 10^11^ M), but P104 micelles showed an important cytotoxicity (IC_50_ = 82.32 µM).

Based on these results the concentration of the copolymer in drug-loaded P104 micelles was kept below 10 µM. DOXO is a cytotoxic antineoplastic drug that inhibits topoisomerase II and induces ROS-associated damage in cancer cells [52,53,54]. IC_50_ values for DOXO in HeLa cells have been reported covering three orders of magnitude. For instance, Lalitha et al. reported an IC_50_ = 24 µM after 48 h in HeLa cells for DOXO [55], while Benyettou et al. reported an IC_50_ = 2.4 µM for the same cell line and incubation time [56].

Compared to previously reported values of IC_50_ of DOXO for HeLa cells after 48 h, we observed that P104_DOXO-B and F127_DOXO-B had an IC_50_ of 0.0327 and 0.0203 µM, respectively, which accounted, at least, for a 70-fold and 120-fold lower IC_50_, respectively, indicating an increased inhibition of viability of DOXO-B when entrapped in micelles, probably originating from the effective incorporation of the nanocarriers inside cells and/or the avoidance of drug-efflux mechanisms [57].

Nevertheless, it is important to bear in mind that viability or cytotoxicity quantification can be influenced by different experimental parameters such as the selected color-generating reagent, cell-culture conditions, number of passages of cells, etc. On the other hand, the IC_50_ of DOCE, a taxane-based anticancer drug, has also been evaluated in different cancer cell lines, including HeLa cells with very dissimilar reported values. For instance, Altamimi et al. reported an IC_50_ = 9.65 µM in HeLa cells determined by the MTT assay [58].

Muzammil Afzal et al. reported an IC_50_ = 41.71 ± 1.67 nM for HeLa cells after 48 h (DOCE was dissolved in RPMI 1640 containing 1% dimethyl sulfoxide (DMSO), also used as control) [59]; and Balcer-Kubizcek reported, for the commercial version of DOCE known as Taxotere, an IC_50_ = 0.3 nM after 24 h in HeLa cells (the drug was previously dissolved in DMSO, stored until used and diluted in culture medium when tested; the authors used mock controls and set a final concentration of DMSO in drug or mock exposure groups ≤ 0.05%) [60].

The IC_50_ values of DOCE in P104 and F127 micelles were 0.00126 and 0.00003 µM (Table 6), respectively. The IC_50_ of DOCE in P104 micelles for HeLa cells was decreased, at least when compared to the data of Altamimi and Muzammil Afzal, and notably, the cytotoxic effect of the drug was increased at least tenfold when entrapped in F127 polymeric-based micelles, as compared to the previous reports above referenced. It is important to remark that due to the solubilizing properties of both copolymers, no DMSO was needed for the solubilization of DOCE when entrapped in the micellar formulations.

Regarding the synergistic effect of DOCE-DOXO, it has previously been evaluated by Tsakalozou et al. in hormone refractory prostate cancer cells for a wide number of combination ratios [61], and using the combination index (*CI*) to determine the synergy (*CI* < 0.9), additivity (0.9 < *CI* < 1.1) and antagonism (*CI* > 1.1) of the combined treatment, where the CI is determined by:(9)$$CI=\frac{\left[DOCE\right]}{{\left[DOCE\right]}_{x}}+\frac{\left[DOXO\right]}{{\left[DOXO\right]}_{x}}+\frac{\left[DOCE\right]\left[DOXO\right]}{{\left[DOCE\right]}_{x}{\left[DOXO\right]}_{x}}$$
where [*DOCE*] and [*DOXO*] are drug concentrations of DOCE and DOXO in combination, inhibiting *x*% of cell viability. [*DOCE*]*_x_* and [*DOXO*]*_x_* are the doses of DOCE and DOXO alone, respectively, inhibiting *x*% of cell viability.

In this work, the authors identified optimal combinations with DOCE concentrations below its IC_50_ and DOXO concentrations of two- to fourfold above their individual IC_50_. Our present results show that the P104_DOCE-DOXO-B system had a strong synergy (CI = 0.1694) calculated for x = 50%. Conversely, the F127_DOCE-DOXO-B system showed a strong antagonism (CI = 942) due to the extremely low IC_50_ of the F127_DOCE micelles; thus, this individual drug micellar solution would probably show a better in vivo performance than the nanoformulation combination.

## 4. Conclusions

Using Pluronic F127 and P104 micellar solutions to solubilize antineoplastic drugs resulted in the following improvements: *(a)* The water solubility of DOCE and DOXO-B drugs was significantly augmented. The solubility of DOCE increased by a factor of 43 and that of DOXO-B by a factor of 21 when Pluronic F127 micelles were used as nanocarriers; when using a Pluronic P104 micellar solution, the solubility of DOCE increased by a factor of 147 and that of DOXO-B by a factor of 61; *(b)* As most of the drug is encapsulated inside the polymeric micelles, the loss of drug would be minimal before reaching the diseased tumoral area; *(c)* Due to the hydrophilic micellar shell, the clearance of the drug by the reticuloendothelial system is prevented; *(d)* After 40 h, the amount of drug release from the micelles still continues, which allows its accumulation in tumors; in addition, it was observed that the release rate is faster at acidic pH (5.5) than at approximately neutral pH, and that the drug release follows the Korsmeyer–Peppas model.

Nowadays, several research groups are currently revisiting multidrug therapies against cancer, using drugs with non-overlapping action mechanisms. However, there is still uncertainty if the combination of drugs in a single carrier can outperform the free or individual drug carriers. In this work, DOXO and DOCE drugs loaded in Pluronic F127 and P104 micelles showed lower IC_50_ values than those reported by other researchers using polymeric nanoparticles, dendrimers or lipid nanoemulsions as alternative carriers, which highlights the importance of the nanoformulation to be used.

Furthermore, the P104_DOCE-DOXO-B micellar system showed synergism by having a higher growth inhibitory capacity than P104 micelles encapsulating either of the single drugs. However, no synergism was observed in the F127_DOCE-DOXO-B micellar system, whereas the F127_ DOCE system had the highest cell-growth-inhibition capacity. Globally, such positive results open the panorama and provide opportunities to explore a broader range of combination ratios by the encapsulation of multiple drugs in a nanomedical formulation.

## Figures and Tables

**Figure 1 polymers-15-02249-f001:**
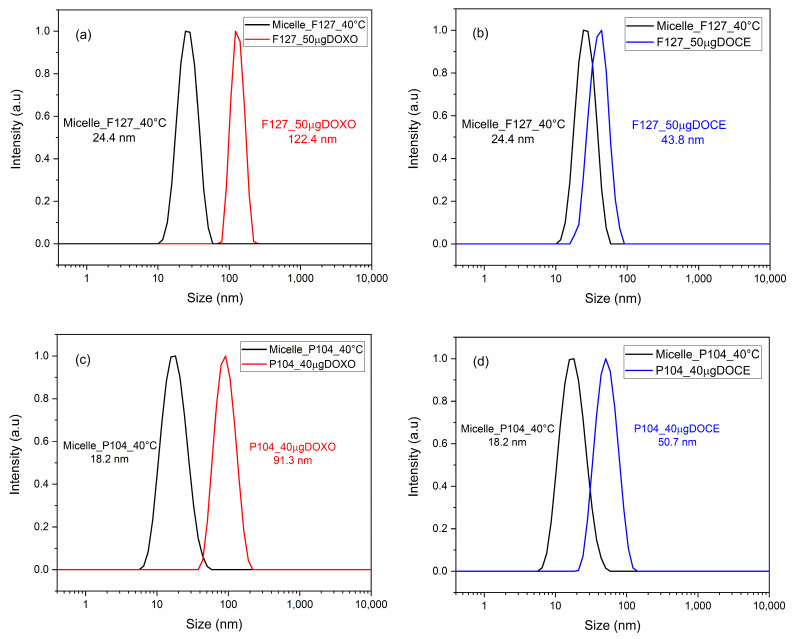
Size distribution at 40 °C of (**a**) unloaded and DOXO-loaded F127 polymeric micelles, (**b**) unloaded and DOCE-loaded F127 polymeric micelles, (**c**) unloaded and DOXO-loaded P104 polymeric micelles and (**d**) unloaded and DOCE-loaded P104 polymeric micelles.

**Figure 2 polymers-15-02249-f002:**
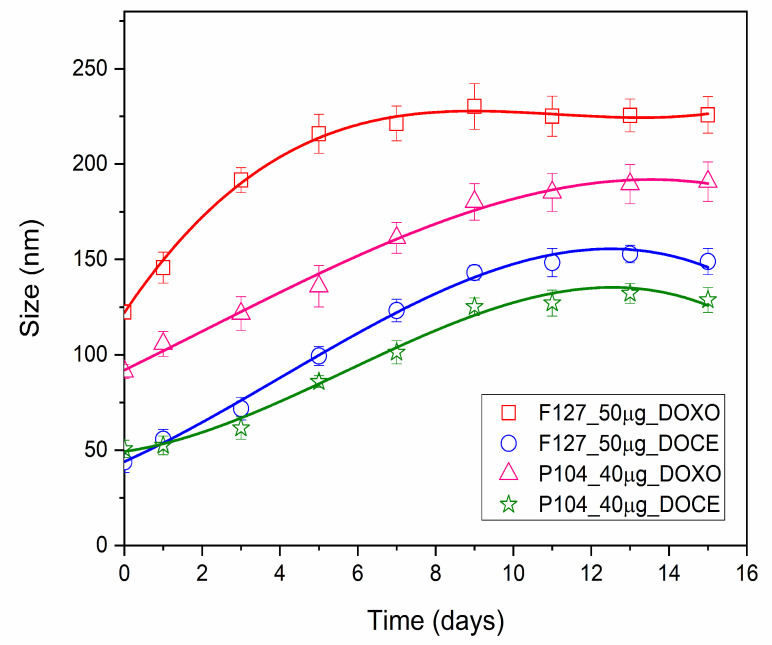
Size stability of Pluronic F127 and P104 micelles loaded with DOCE and DOXO drugs at 40 °C.

**Figure 3 polymers-15-02249-f003:**
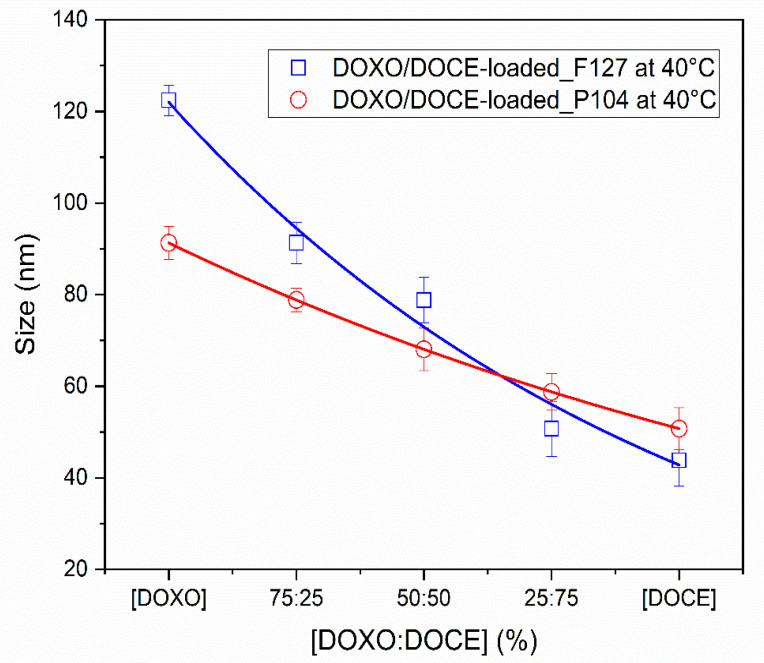
Hydrodynamic diameter evolution for dual DOCE/DOXO-loaded (
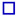
) F127 and (
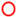
) P104 polymer micelles.

**Figure 4 polymers-15-02249-f004:**
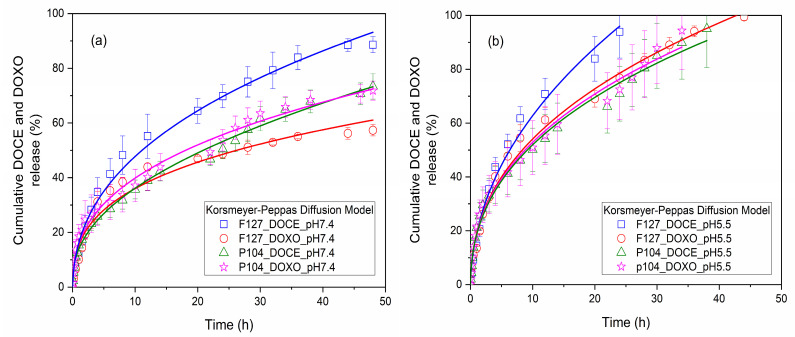
In vitro DOCE and DOXO release from loaded F127 and P104 polymeric micelles at 37 °C (**a**) pH 7.4 and (**b**) pH 5.5.

**Figure 5 polymers-15-02249-f005:**
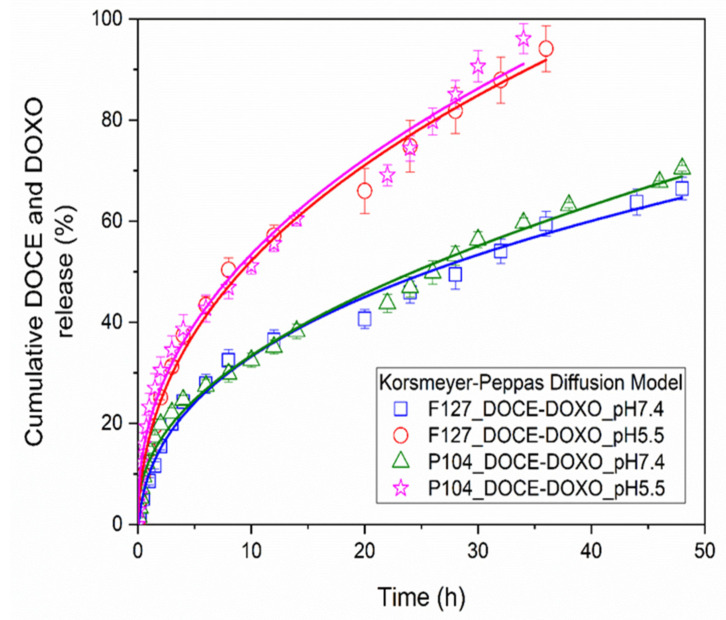
In vitro DOCE-DOXO release from loaded F127 and P104 polymeric micelles at 37 °C as a function of pH solution.

**Figure 6 polymers-15-02249-f006:**
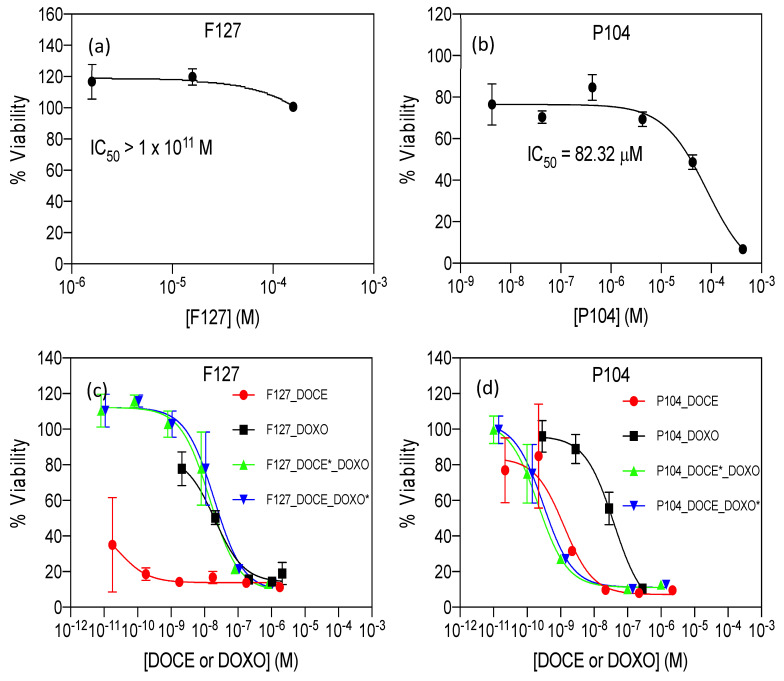
Dose-response curves of (**a**) F127 empty micelles, (**b**) P104 empty micelles, (**c**) F127 drug-loaded micelles, (**d**) P104 drug-loaded micelles. The viability inhibition capacity of micelles was evaluated in HeLa cells after 48 h of exposure under standard culture conditions (37 °C and 5% CO_2_). Data is reported as the average of three replicates ± standard deviation.

**Table 1 polymers-15-02249-t001:** Parameters obtained in the loading process of individual docetaxel and doxorubicin in the micellar solutions of Pluronic F127 and P104.

* D/C (*w*/*w*%)	(μg)		*D.L.* (*wt*%)	*E.E.* (*wt*%)	*S_cp_* (mg·g^−1^)	(μg)		*D.L.* (*wt*%)	*E.E.* (*wt*%)	*S_cp_* (mg·g^−1^)
	DOCE loading (PEO_100_PPO_65_PEO_100_)					DOXO loading (PEO_100_PPO_65_PEO_100_)				
0.05	6.45	±0.93	0.031	64.5	0.313	6.02	±0.91	0.030	60.2	0.303
0.10	11.82	±0.99	0.058	59.1	0.583	10.42	±1.43	0.049	52.1	0.495
0.15	16.97	±0.75	0.088	56.6	0.876	14.34	±1.98	0.068	47.8	0.685
0.20	21.76	±0.54	0.115	54.4	1.147	17.31	±1.90	0.085	43.3	0.849
0.25	28.42	±0.90	0.142	56.8	1.424	22.90	±1.74	0.114	45.8	1.145
0.30	27.61	±0.81	0.141	46.0	1.408	24.50	±1.29	0.117	40.8	1.171
	DOCE loading (PEO_27_PPO_61_PEO_27_)					DOXO loading (PEO_27_PPO_61_PEO_27_)				
0.02	8.50	±0.02	0.017	85.0	0.170	8.75	±0.49	0.017	87.5	0.175
0.04	15.77	±0.40	0.032	78.9	0.315	15.22	±0.74	0.030	76.1	0.304
0.06	21.61	±1.42	0.043	72.0	0.432	21.75	±0.63	0.043	72.5	0.435
0.08	35.40	±1.24	0.071	88.5	0.710	30.41	±1.53	0.061	76.0	0.609
0.10	40.41	±0.57	0.081	80.8	0.808	31.39	±1.14	0.063	62.8	0.628
0.12	42.15	±0.63	0.084	70.3	0.843	32.55	±1.48	0.065	54.2	0.651

* D/C is the drug/copolymer ratio.

**Table 2 polymers-15-02249-t002:** Parameters obtained for the DOCE/DOXO dual charge in the micellar solution of Pluronic F127 and P104.

Copolymers	[DOXO/DOCE] (*w*/*w*%)	DOXO (μg)		DOCE (μg)		*D.L.* (*wt*%)	*E.E.* (*wt*%)	*S_cp_* (mg·g^−1^)
PEO_100_PPO_65_PEO_100_ (2 *wt*%)	100/0	22.90	±1.74	0.00	0.00	0.046	45.8	1.141
	75/25	16.77	±2.08	6.69	±0.38	0.047	46.9	1.173
	50/50	11.76	±0.91	12.86	±1.09	0.049	49.2	1.231
	25/75	4.89	±0.1	19.35	±1.43	0.049	48.5	1.214
	0/100	0.00	0.00	28.42	±0.90	0.057	56.8	1.424
PEO_27_PPO_61_PEO_27_ (5 *wt*%)	100/0	30.41	±1.53	0.00	0.00	0.076	76.0	0.609
	75/25	22.33	±0.67	8.36	±0.47	0.077	76.7	0.614
	50/50	15.34	±0.71	16.08	±1.37	0.079	78.6	0.628
	25/75	6.55	±0.49	24.22	±1.79	0.077	76.9	0.615
	0/100	0.00	0.00	35.40	±1.24	0.089	88.5	0.712

**Table 3 polymers-15-02249-t003:** Coefficients of DOCE release from single-loaded F127 and P104 polymeric micelles.

Diffusion Models		F127_DOCE				P104_DOCE			
		pH 7.4		pH 5.5		pH 7.4		pH 5.5	
Higuchi	*k*	14.05	±0.92	18.96	±1.25	10.90	±0.77	15.36	±2.32
	R^2^	0.973		0.987		0.985		0.985	
	MSE	26.87		25.45		8.43		17.53	
	MSC	3.59		4.17		4.12		4.08	
Korsmeyer–Peppas	*k_1_*	17.33	±3.61	21.11	±2.87	13.53	±1.84	18.41	±3.99
	*n*	0.44	±0.04	0.47	±0.02	0.44	±0.02	0.45	±0.04
	R^2^	0.985		0.990		0.993		0.992	
	MSE	15.89		21.33		3.67		9.61	
	MSC	3.90		4.24		4.83		4.60	
Peppas–Sahlin	*k_2_*	15.60	±3.37	18.88	±1.62	12.82	±1.86	17.90	±3.72
	*k_3_*	−0.68	±0.31	−0.65	±0.17	1.08	±0.12	0.79	±1.01
	*m*	0.59	±0.03	0.60	±0.04	0.37	±0.01	0.41	±0.04
	R^2^	0.993		0.995		0.994		0.993	
	MSE	7.72		10.11		3.47		9.23	
	MSC	4.57		4.92		4.86		4.60	

**Table 4 polymers-15-02249-t004:** Coefficients of DOXO release from single-loaded F127 and P104 polymeric micelles.

Diffusion Models		F127_DOXO				P104_DOXO			
		pH 7.4		pH 5.5		pH 7.4		pH 5.5	
Higuchi	*k*	9.74	±0.04	15.85	±0.52	11.28	±0.77	16.02	±2.51
	R^2^	0.876		0.979		0.981		0.984	
	MSE	48.95		25.96		10.81		22.07	
	MSC	1.84		3.75		3.14		4.06	
Korsmeyer–Peppas	*k_1_*	16.20	±1.21	19.89	±2.39	16.07	±5.01	17.89	±5.10
	*n*	0.34	±0.02	0.43	±0.03	0.40	±0.08	0.47	±0.06
	R^2^	0.954		0.990		0.987		0.990	
	MSE	19.36		12.86		7.75		14.57	
	MSC	2.72		4.32		3.98		4.37	
Peppas–Sahlin	*k_2_*	14.84	±1.76	19.17	±2.21	16.28	±5.51	17.06	±5.74
	*k_3_*	−0.99	±0.26	−0.73	±0.26	−0.15	±0.72	1.38	±1.20
	*m*	0.54	±0.05	0.52	±0.01	0.40	±0.05	0.40	±0.01
	R^2^	0.981		0.992		0.988		0.991	
	MSE	8.30		10.42		7.59		12.90	
	MSC	3.47		4.47		4.08		4.45	

**Table 5 polymers-15-02249-t005:** Coefficients for DOCE and DOXO release from dual-loaded F127 and P104 polymeric micelles.

Diffusion Models		F127_DOCE-DOXO				P104_DOCE-DOXO			
		pH 7.4		pH 5.5		pH 7.4		pH 5.5	
Higuchi	*k*	9.75	±0.42	15.63	±0.65	10.18	±0.24	16.81	±0.55
	R^2^	0.985		0.990		0.985		0.986	
	MSE	7.02		11.75		7.16		17.23	
	MSC	3.99		4.47		4.06		4.19	
Korsmeyer–Peppas	*k_1_*	11.06	±0.92	17.91	±0.38	12.24	±0.93	19.18	±1.86
	*n*	0.46	±0.02	0.46	±0.01	0.44	±0.02	0.45	±0.02
	R^2^	0.989		0.994		0.991		0.991	
	MSE	5.73		8.13		4.38		11.63	
	MSC	4.09		4.78		4.53		4.50	
Peppas–Sahlin	*k_2_*	11.06	±0.92	17.55	±0.55	11.59	±1.36	17.47	±1.13
	*k_3_*	−0.22	±0.09	−0.23	±0.37	0.97	±0.54	2.36	±0.69
	*m*	0.50	±0.01	0.49	±0.06	0.38	±0.02	0.37	±0.02
	R^2^	0.989		0.994		0.992		0.993	
	MSE	5.57		7.87		4.05		10.20	
	MSC	4.16		4.83		4.56		4.60	

**Table 6 polymers-15-02249-t006:** IC_50_ of micellar systems evaluated in HeLa cells after 48 h of co-culture obtained by an (inhibitor) vs. response three parameter model.

Micellar System	IC_50_ (µM)	R^2^
P104 empty	82.32	0.943
P104_DOCE	0.00126	0.888
P104_DOXO	0.03270	0.975
P104_DOCE-DOXO	0.0002 ^a^ 0.0003 ^b^	0.966 ^a^ 0.966 ^b^
F127 empty	>1 × 10^11^	-
F127_DOCE	0.00003	0.487
F127_DOXO	0.02030	0.956
F127_DOCE-DOXO	0.0144 ^a^ 0.0195 ^b^	0.959 ^a^ 0.959 ^b^

^a^ IC_50_ based on [DOCE]; ^b^ IC_50_ based on [DOXO].

## Data Availability

The data generated from this research are available from the authors.

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
