# Peer review of "Pluronic F127 and P104 Polymeric Micelles as Efficient Nanocarriers for Loading and Release of Single and Dual Antineoplastic Drugs"

_polymers, 2023, doi:10.3390/polym15102249_

Round 1

Reviewer 1 Report

The paper “Pluronics® F127 and P104 Polymeric Micelles as Efficient Nanocarriers for Loading and Release of Single and DualAntineoplastic Drugsis focused to design and study the potential application of biodegradable and biocompatible polymeric micelles formed by Pluronics® F127 and P104 as nanocarriers of the antineoplastic drugs.

The document is interesting, well organised, but a check of the language and grammar should always be done.

Have they been characterised from the point of view of stability over time? It would be good to add this study or an accelerated stability assay to better characterise the micelles from a technological point of view.

The figures have good resolution, and the conclusions also frame the work well.

control over language, both grammatical and orthographic.

Author Response

The paper “Pluronics® F127 and P104 Polymeric Micelles as Efficient Nanocarriers for Loading and Release of Single and DualAntineoplastic Drugs” is focused to design and study the potential application of biodegradable and biocompatible polymeric micelles formed by Pluronics® F127 and P104 as nanocarriers of the antineoplastic drugs.

Point 1: The document is interesting, well organised, but a check of the language and grammar should always be done.

Response 1: We appreciate your comments. A language and grammar check was performed.

Point 2: Have they been characterised from the point of view of stability over time? It would be good to add this study or an accelerated stability assay to better characterise the micelles from a technological point of view.

Response 2: Attending your suggestion, we are including a plot of the stability of Pluronics® F127 and P104 DOCE and DOXO-loaded micelles following the evolution of micellar size over time (Figure 2 in the second version of the article). In such Figure, it can be observed that the size of the micelles increases in the first days, before it becomes fairly constant after the eleventh day. Similar behavior was observed by Villar et al., who reported an increase in size in poly(butylene oxide)-poly(ethylene oxide)-poly(butylene oxide) polymeric micelles loaded with DOCE/DOXO. The size increase occurs by a structural rearrangement of the micelles [7]. The plot explanation in the article is included on lines 359 to 369.

Figure 2. Size stability of Pluronics® F127 and P104 micelles loaded with DOCE and DOXO drugs at 40 °C. Graphic added on lines 392.

Point 3: The figures have good resolution, and the conclusions also frame the work well.

Response 3: The authors agree with your comment

Reviewer 2 Report

Edgar B. Figueroa-Ochoa and his team presented a manuscript mainly describing Pluronics F127 and P104-based polymeric micelles as efficient nanocarriers for loading doxorubicin and docetaxel. The team analyzed the drug release profiles using various mathematical models. The authors also calculated IC50 values of polymeric micelles after exposing them to HeLa cells. 

Overall, the manuscript is well-conceived and well-written; hence it is worth publishing in MDPI Polymers. Address the following minor issues. 

1.    Redundant acronyms were observed.                                                                           For example: Lines 58 and 62: Polyethylene oxide (PEO) and polypropylene oxide (PPO) 

2.    Configural descriptors must be shown in small capital letters.                                 Line 102: poly(γ-benzyl-L-glutamate) --> poly(γ-benzyl-l-glutamate) 

3.    Discuss why 40 ℃ was used for DLS analysis. 

Author Response

Edgar B. Figueroa-Ochoa and his team presented a manuscript mainly describing Pluronics F127 and P104-based polymeric micelles as efficient nanocarriers for loading doxorubicin and docetaxel. The team analyzed the drug release profiles using various mathematical models. The authors also calculated IC50 values of polymeric micelles after exposing them to HeLa cells.

Overall, the manuscript is well-conceived and well-written; hence it is worth publishing in MDPI Polymers. Address the following minor issues.

Response: We appreciate your comments.

Point 1: Redundant acronyms were observed. For example: Lines 58 and 62: Polyethylene oxide (PEO) and polypropylene oxide (PPO) .

Response 1: Redundant acronyms were removed specifically from lines 62 and 63.

Point 2: Configural descriptors must be shown in small capital letters. Line 102: poly(γ-benzyl-L-glutamate) --> poly(γ-benzyl-L-glutamate)

Response 2: The configural descriptors of line 102 were modified.

Point 3: Discuss why 40 ℃ was used for DLS analysis.

 Response 3: The formation of micellar structures of Pluronic® F127 and P104 in aqueous solution is a function of the critical micellar concentration (CMC) and the critical micellar temperature (CMT). Specifically for these materials, the CMT ranges from 18 to 36 °C, the temperature range at which micelles begin to form in dilute aqueous solution [40]. It is for this reason that we decided to use the temperature of 40 °C in the DLS experiments, to ensuring complete micelle formation. Discussion on lines 339 to 340.

Reviewer 3 Report

1, The authors adopted commercial-available polymers to encapsulate marketed drugs. What are the essential contribution to the field of polymers? I am afraid the current status cannot meet the requirements of the journal polymers.

2, An illustration is missing to describe the main contents.

3, Besides increasing the solubility, another important aim of micellization is to improve the in vivo pharmacokinetics, for example, to target tumor via EPR effect. However, the authors unfortunately failed to carry on any in vivo experiments.

4, The index of cooperation of two drugs in killing tumor cells in a single system should be provided.

Author Response

Point 1: The authors adopted commercial-available polymers to encapsulate marketed drugs. What are the essential contribution to the field of polymers? I am afraid the current status cannot meet the requirements of the journal polymers.

Response 1: In this work, chemotherapeutic drugs and commercial triblock copolymers (FDA approved) were used to be able to use them in real life for cancer treatment. Furthermore, this work is a contribution in line with several  of the aims and scope of the Polymers Journal, developing a new nanocarrier-drug formulation and modeling drug release kinetics, in conjunction with characterization and analysis of  the polymeric  system used, for its application in the medical area.

Basically, here we report a drug carrier system using a biodegradable and biocompatible micellar solution loaded with the drugs docetaxel (DOCE) and doxorubicin (DOXO), along with the drug release profiles at different pH, showing that an efficient drug delivery nanocarrier system was obtained. These formulations showed lower IC50 values than those reported by other researchers that used polymeric nanoparticles, dendrimers, or liposomes as alternative drug carriers, indicating that with our system, a lower drug concentration is needed to decrease cell viability by 50%.

Point 2: An illustration is missing to describe the main contents.

Response 2: The figure placed as a graphical abstract illustrates the main contents of the article.

Point 3: Besides increasing the solubility, another important aim of micellization is to improve the in vivo pharmacokinetics, for example, to target tumor via EPR effect. However, the authors unfortunately failed to carry on any in vivo experiments.

Response 3: The analysis of assays in biological models or in vivo experiments were not contemplated within the scope of this project.

Point 4: The index of cooperation of two drugs in killing tumor cells in a single system should be provided.

Response 4: We appreciate your accurate observation. We have conceived a combination chemotherapy in a single device; however, we consider that the appoach proposed by the reviewer is very interesting, that it would yield different results and probably a better combination index, nonetheless, we have not performed the combination treatment with single-drug micelles, and we are not able to make such calculation. Based on your recommendation, we will explore this approach in futher publications.

Reviewer 4 Report

The research work focuses on developing biodegradable and biocompatible polymeric micelles using Pluronics® F127 and P104 as nanocarriers of the antineoplastic drugs docetaxel and doxorubicin. The formulated product showed lower IC50 values than another type of nanoformulation recently developed products. The revision and addition of data (Zeta potential, stability and Solvent quantity by GC) and justification are required as follows to improve the manuscript.

1.      Line 138: Specify the quantity of solvent. dichloromethane used or its QS?

2.      Line 141: Is it correct, 1 mL of deionized water?

3.      Line 143: stirred for five days, please specify the conditions, speed, stirrer type and temperature.

4.      Line 148: protect them from light,  is drug light sensitive?

5.      Line 184: PBS buffer medium at pH of 5.5 184 and 7.4; Why are two buffer systems used for drug release?

6.      Line 258: Dichloromethane, was evaporated. Please provide, GC data, to confirm that it's not present in the product and if present that it's within limits.

7.      Please provide zeta potential data, along with particle size and also discuss its effect on stability.

8.      Need to confirm the stability of the prepared product.

Author Response

The research work focuses on developing biodegradable and biocompatible polymeric micelles using Pluronics® F127 and P104 as nanocarriers of the antineoplastic drugs docetaxel and doxorubicin. The formulated product showed lower IC50 values than another type of nanoformulation recently developed products. The revision and addition of data (Zeta potential, stability and Solvent quantity by GC) and justification are required as follows to improve the manuscript.

Response: We appreciate your comments.

Point 1: Line 138: Specify the quantity of solvent. dichloromethane used or its QS?

Response 1: Solutions of the DOCE or DOXO drugs are prepared in dichloromethane at a ratio of 200 µg/mL, then 50 to 300 µL are added to 20 mg of F127 or to 50 mg of P104 to form the drug-loaded micellar solutions reported in Table 1. Quantity of solvent dichloromethane specified in lines 138-140.

Point 2: Line 141: Is it correct, 1 mL of deionized water?

Response 2: Effectively, 1 mL of deionized water was used.

Point 3: Line 143: stirred for five days, please specify the conditions, speed, stirrer type and temperature.

Response 3: Once the drug-encapsulated micellar system is obtained, the vials are keep uncapped, under magnetic stirring at 250 rpm and room temperature of 25 °C for five days to evaporating the organic solvent. Conditions specified in lines 142-144.

Point 4: Line 148: protect them from light, is drug light sensitive?

Response 4: Effectively, the doxorubicin is high sensitivity to natural or artifiial light

Point 5: Line 184: PBS buffer medium at pH of 5.5 184 and 7.4; Why are two buffer systems used for drug release?

Response 5: Two PBS buffer medium at different pH were used, trying to simulate intestinal fluid (pH 5.5) and parenteral administration (pH 7.4), to analyze the effect on the kinetic profile of drug-loaded micellar systems. Specified in line 186.

Point 6: Line 258: Dichloromethane, was evaporated. Please provide, GC data, to confirm that it's not present in the product and if present that it's within limits.

Response 6: The amount of dichloromethane (highly volatile) used to prepare the micellar solutions was very small, 50 to 300 µL of drug/dichloromethane solution (200 µg·mL-1) are added to each Pluronic®. The mixtures are kept uncapped, under magnetic stirring at 250 rpm and room temperature of 25 ° C for five days to evaporating the organic solvent. Unfortunately performing GC analysis is beyond the scope of this project, so we do not have GC data to confirm it.

Point 7: Please provide zeta potential data, along with particle size and also discuss its effect on stability.

Response 7: The stability of DOCE and DOXO loaded in micelles of Pluronics® F127 and P104 determined by monitoring the micellar size evolution.

Point 8: Need to confirm the stability of the prepared product.

Response 8: Attending your suggestion, we are including a plot of the stability of Pluronics® F127 and P104 DOCE and DOXO-loaded micelles following the evolution of micellar size over time (Figure 2 in the second version of the article). In such Figure, it can be observed that the size of the micelles increases in the first days, before it becomes fairly constant after the eleventh day. Similar behavior was observed by Villar et al., who reported an increase in size in poly(butylene oxide)-poly(ethylene oxide)-poly(butylene oxide) polymeric micelles loaded with DOCE/DOXO. The size increase occurs by a structural rearrangement of the micelles [7]. The plot explanation in the article is included on lines 359 to 369.

Figure 2. Size stability of Pluronics® F127 and P104 micelles loaded with DOCE and DOXO drugs at 40 °C. Graphic added on lines 392.

Round 2

Reviewer 3 Report

The authors basically adressed the comments in the first round review, and improved the manuscript to a higher level. More ref, especially reviews could be added to help discuss the dual-loading and release in a singale system could be more helpful, suhc as 10.1002/adhm.202300138,10.1016/j.addr.2023.114773 and 10.1016/j.cclet.2022.05.032. In general, I am happy to see the advancements of this work.

Reviewer 4 Report

All the answers were satisfactory and corrections are done accordingly.